# Does Urbanization Improve Industrial Water Consumption Efficiency?

**Bingquan Liu [1,2,*], Yongqing Li [1], Rui Hou [1] and Hui Wang [1,2]**

[1]   School of Economics & Management, China University of Petroleum (East China), Qingdao 266580, China; liyongqing@163.com (Y.L.); hourui@163.com (R.H.); Wanghui@126.com (H.W.)
[2]   Institute for Energy Economics and Policy, China University of Petroleum (East China), Qingdao 266580, China
*   Correspondence: liubq@upc.edu.cn or jgliubingquan@126.com

**Abstract:** Although some studies have focused on the logical connection between industrial water consumption in the industrial economic development (IED) and industrial wastewater treatment (IWT) stages, the master–slave game relationship between these stages has not been considered. This study selected panel data from 30 provinces in China from 2011 to 2015, divided these provinces into IED- and IWT-dominated regions, and developed a two-stage data envelopment analysis (DEA) model based on the master–slave game relationship between the IED and IWT stages. In addition, a regression model based on the Simar–Wilson approach was constructed to reveal the effects of urbanization on industrial water consumption efficiency. The results show that the industrial water consumption efficiency in China slightly fluctuated from 2011 to 2015, and there was no significant efficiency improvement. The efficiency of the IED stage was generally higher than that of the IWT stage, and the efficiency gap between stages was smaller in IED-dominated regions than in IWT-dominated regions. Urbanization has different effects on industrial water consumption efficiency, and the same factor can have significantly different effects in different regions. Some policy implications are proposed for the different types of regions.

**Keywords:** industrial water consumption efficiency; urbanization; master–slave game

## 1. Introduction

Although the development of industry in China has achieved great success since the reform and expansion plans were implemented, industrial water consumption and wastewater disposal issues have become increasingly serious. In 2016, the proportion of water consumption by industries in China reached 21.7%, and the total amount of wastewater discharged reached 71.10 billion tons [1]. At the global scale, China is not a water-rich country, and rapid urbanization has occurred in recent years. This urbanization has promoted industrial development [2], which has increased industrial water consumption and wastewater discharge. Therefore, the effects of urbanization on industrial water consumption efficiency must be analyzed, and the efficiency of water use should be improved with new methods during this period of rapid urbanization in China.

In recent years, the efficiency of industrial water consumption has been studied by scholars. There are two important issues related to this topic: determining how to scientifically evaluate industrial water consumption efficiency and identifying the key factors that affect industrial water consumption efficiency to develop reasonable policies.

For the first issue, studies of industrial water consumption efficiency have been gradually refined, and different types of data envelopment analysis (DEA) models have been used widely. In one study, water consumption efficiency was defined as the economic output per unit of water consumption [3].

However, these studies did not consider the corresponding environmental pollution [4,5]. Hu et al. [6] proposed a total-factor efficiency evaluation model that included both desired and undesired outputs. This approach was more consistent with the actual process of industrial water consumption compared with other models. Bian et al., Sala-Garrido et al., and Deng et al. [7–9] performed subsequent studies based on this concept.

Although the abovementioned studies fully considered the desired and undesired outputs, they still regarded the process of industrial water consumption as a "black box" and did not consider the process of industrial water consumption in detail. Specifically, the process of industrial water consumption can be divided into several stages. Färe and Grosskopf [10] proposed the first network DEA model. Later, many studies began to divide the industrial water consumption process into two stages, namely, the water consumption stage and wastewater treatment stage, to truly reflect the actual process of industrial water consumption. Since then, many studies have applied the same concept to obtain detailed results [11–13]. Most studies, though, have focused on the hierarchical relationship between the industrial water consumption stage and wastewater treatment stage. However, there are different development goals in different regions [14]. For example, some regions would benefit most from rapid economic development and can focus less on wastewater treatment, while others must focus on wastewater treatment and slow the rate of economic development. Therefore, a master–slave game relationship exists between the industrial water consumption stage and wastewater treatment stage, and this relationship is often not considered.

For the second issue, many studies have focused on industrial structure, economic development, technical progress, and expansion policies to determine the key factors that influence industrial water consumption [6,9,15–21]. However, most of these factors are affected by the rapid urbanization; thus, urbanization is the main factor that has influenced the efficiency of industrial water consumption in China. In this context, it is important to determine the effects of urbanization on the efficiency of industrial water consumption. In addition, most previous studies analyzed the effects of various influential factors on overall efficiency, but such effects should be separately considered for both stages of industrial water consumption in different types of regions.

In addition, many studies have used Tobit to determine the key factors that influence industrial water consumption because industrial water consumption efficiency is restricted to the interval zero and one [9,17,21–23]. However, the environmental variables used in Tobit models are probably correlated with the efficiency scores calculated in DEA models, which may lead to the inconsistency problems of estimators [24]. Another problem is that the true efficiency score is not observed directly but is empirically estimated [25]. So, many studies began to focus on the Simar–Wilson approach, which is based on a double bootstrap and can deal with these challenges.

In summary, although the efficiency of industrial water consumption has been broadly studied, additional research is needed to fill specific knowledge gaps: (1) Provinces have significantly different development goals because the economic development levels and water resource endowments in different provinces are imbalanced in China, partly due to the effects of urbanization [26,27]. Some provinces may require more industrial economic development (IED) and less industrial wastewater treatment (IWT). In this case, economic development is the "master" goal and wastewater treatment is the "slave" goal in the industrial water consumption process. In other regions, the opposite may be true. Therefore, it is necessary to construct a model that considers this master–slave relationship between IED and IWT to evaluate industrial water consumption efficiency and obtain objective results. (2) With rapid urbanization in China, the population distribution, industrial structure, spatial structure, and consumption structure of provinces have changed, and these changes have affected industrial water consumption efficiency. Therefore, the effect of urbanization on industrial water consumption efficiency must be analyzed for different stages of industrial water consumption.

Based on the above discussion, this paper focuses on the following objectives: (1) According to the regional development characteristics, 30 provinces in China are divided into two classes: IED-dominated provinces and IWT-dominated provinces. Then, a two-stage efficiency evaluation

model is established based on a master–slave game, and the industrial water consumption efficiency of 30 provinces in China is evaluated. Finally, the differences between the two stages in different provinces are investigated. (2) Based on statistical data from 2010 to 2015 in China, a regression model based on the Simar–Wilson approach is constructed to determine how urbanization has affected the two stages and overall efficiency of industrial water consumption.

## 2. Methodology

### 2.1. Two-Stage Efficiency Evaluation Model

Suppose there are $n$ decision making units ($DMU_s$), which represent provinces that should be evaluated. In the process of industrial water consumption, each $DMU$ uses water and other resources to generate an industrial GDP, and some wastewater is produced. Then, various types of equipment and technology are used to treat the wastewater, which saves water resources and protects the environment. Therefore, the process of industrial water consumption can be divided into two stages. The first stage aims to increase desired outputs and decrease undesired outputs, and the second stage aims to increase the amount of treated wastewater. Therefore, we define the first stage as the IED stage and the second stage as the IWT stage. The corresponding process is shown as Figure 1.

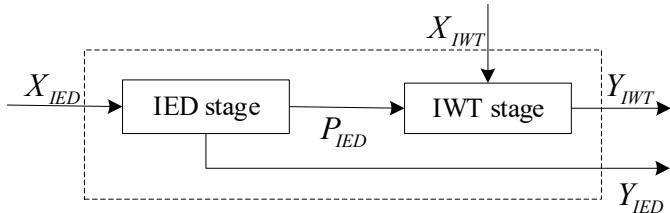

**Figure 1.** Two stages of industrial water consumption and treatment.

For $DMU_0$, $E_0^{IED} = \left\{ (X_{IED_0}, Y_{IED_0}, P_{IED_0}) \middle| X_{IED_0} can\ produce(Y_{IED_0}, P_{IED_0}) \right\}$ describes the variable relations in the IED stage. In this case, $X_{IED_0}$ is the input, $Y_{IED_0}$ is the final desired output, and $P_{IED_0}$ is the undesired output, which is also one of the inputs of the IWT stage. $E_0^{IWT} = \left\{ (X_{IWT_0}, Y_{IWT_0}) \middle| X_{IWT_0} can\ produce\ Y_{IWT_0} \right\}$ describes the relations in the IWT stage. $X_{IWT_0}$ is an input that is necessary for wastewater treatment, such as government investment, and $Y_{IWT_0}$ is the desired output of the IWT stage, such as treated wastewater.

The undesired output conversion function [28,29] and slacks-based measure (SBM) model [30,31] are two important methods for considering undesired outputs. However, the SBM model, a nonradial and nonangle model, considers the influence of slack variables fully and obtains objective results. Therefore, this paper uses the SBM model to address undesirable outputs.

For IED-dominated provinces with IWT assistance, it is necessary to maximize the efficiency of the IED stage first and then maximize the efficiency of the IWT stage under the constraint that the efficiency of the IED stage should remain unchanged. Therefore, the model of the IED stage is defined as shown in Formula (1):

$$
E_0^{IED} = \min \frac{1 - \frac{1}{m_1}\left( \sum\limits_{i=1}^{m_1} \frac{s_i^{X_{IED}}}{X_{IED_{i0}}} \right)}{1 + \frac{1}{s_1 + p_1}\left( \sum\limits_{i=1}^{s_1} \frac{s_i^{Y_{IED}}}{Y_{IED_{i0}}} + \sum\limits_{i=1}^{p_1} \frac{s_i^{P_{IED}}}{P_{IED_{i0}}} \right)}
$$

$$
s.t. \begin{cases} X_{IED}\lambda^{IED} + s^{X_{IED}} = X_{IED_0} \\ Y_{IED}\lambda^{IED} - s^{Y_{IED}} = Y_{IED_0} \\ P_{IED}\lambda^{IED} + s^{P_{IED}} = P_{IED_0} \\ s^{X_{IED}} \geq 0, s^{Y_{IED}} \geq 0, s^{P_{IED}} \geq 0, \lambda^{IED} \geq 0 \end{cases} \tag{1}
$$

where $\lambda^{IED}$ is the weight vector of the IED stage, and $s^{X_{IED}}, s^{Y_{IED}}$, and $s^{P_{IED}}$ are the slack variables of $X_{IED}, Y_{IED}$, and $P_{IED}$, respectively.

If $t = \dfrac{1}{1+\frac{1}{s_1+p_1}\left(\sum\limits_{i=1}^{s_1}\frac{s_i^{Y_{IED}}}{Y_{IED_{i0}}}+\sum\limits_{i=1}^{p_1}\frac{s_i^{P_{IED}}}{P_{IED_{i0}}}\right)}$, then the model can be linearized, as shown in Formula (2):

$$E_0^{IED} = \min t - \frac{1}{m_1}\left(\sum_{i=1}^{m_1}\frac{S_i^{X_{IED}}}{X_{IED_{i0}}}\right)$$

$$s.t.\begin{cases} X_{IED}\Lambda^{IED} + S^{X_{IED}} - tX_{IED_0} = 0 \\ Y_{IED}\Lambda^{IED} - S^{Y_{IED}} - tY_{IED_0} = 0 \\ P_{IED}\Lambda^{IED} + S^{P_{IED}} - tP_{IED_0} = 0 \\ t + \frac{1}{s_1+p_1}\left(\sum\limits_{i=1}^{s_1}\frac{S_i^{Y_{IED}}}{Y_{IED_{i0}}}+\sum\limits_{i=1}^{p_1}\frac{S_i^{P_{IED}}}{P_{IED_{i0}}}\right) = 1 \\ \Lambda^{IED} \geq 0 \\ S^{X_{IED}} \geq 0, S^{Y_{IED}} \geq 0, S^{P_{IED}} \geq 0 \end{cases} \qquad (2)$$

where $S^{X_{IED}} = ts^{X_{IED}}, S^{Y_{IED}} = ts^{Y_{IED}}, S^{P_{IED}} = ts^{P_{IED}}$, and $\Lambda^{IED} = t\lambda^{IED}$.

Then, the evaluation model of the IWT stage can be defined as shown in Formula (3):

$$E_0^{IWT} = \min \frac{1-\frac{1}{p_1+m_2}\left(\sum\limits_{i=1}^{p_1}\frac{s_i^{P_{IED}^*}}{P_{IED_{i0}}}+\sum\limits_{i=1}^{m_2}\frac{s_i^{X_{IWT}}}{X_{IWT_{i0}}}\right)}{1+\frac{1}{s_2}\sum\limits_{i=1}^{s_2}\frac{s_i^{Y_{IWT}}}{Y_{IWT_{i0}}}}$$

$$s.t.\begin{cases} X_{IWT}\lambda^{IWT} + s^{X_{IWT}} = X_{IWT_0} \\ P_{IED}\lambda^{IWT} + s^{P_{IED}^*} = P_{IED_0} \\ Y_{IWT}\lambda^{IWT} - s^{Y_{IWT}} = Y_{IWT_0} \\ \dfrac{1-\frac{1}{m_1}\left(\sum\limits_{i=1}^{m_1}\frac{s_i^{X_{IED}}}{X_{IED_{i0}}}\right)}{1+\frac{1}{s_1+p_1}\left(\sum\limits_{i=1}^{s_1}\frac{s_i^{Y_{IED}}}{Y_{IED_{i0}}}+\sum\limits_{i=1}^{p_1}\frac{s_i^{P_{IED}}}{P_{IED_{i0}}}\right)} = E_0^{IED^*} \\ P_{IED}\lambda^{IWT} = P_{IED}\lambda^{IED} \\ s^{X_{IWT}} \geq 0, s^{P_{IED}^*} \geq 0, s^{Y_{IWT}} \geq 0, s^{X_{IED}} \geq 0, s^{Y_{IED}} \geq 0, s^{P_{IED}} \geq 0 \\ \lambda^{IWT} \geq 0, \lambda^{IED} \geq 0 \end{cases} \qquad (3)$$

where $\lambda^{IED}$ and $\lambda^{IWT}$ are the weight vectors of the IED stage and IWT stage, respectively. Moreover, $\dfrac{1-\frac{1}{m_1}\left(\sum\limits_{i=1}^{m_1}\frac{s_i^{X_{IED}}}{X_{IED_{i0}}}\right)}{1+\frac{1}{s_1+p_1}\left(\sum\limits_{i=1}^{s_1}\frac{s_i^{Y_{IED}}}{Y_{IED_{i0}}}+\sum\limits_{i=1}^{p_1}\frac{s_i^{P_{IED}}}{P_{IED_{i0}}}\right)} = E_0^{IED^*}$ indicates that the IWT stage is constrained by the efficiency of the IED stage. $s^{P_{IED}^*}$ is the slack variable of the IWT stage input $P_{IED}$, and $s^{P_{IED}}$ is the slack variable of the IED stage output $P_{IED}$.

Setting $k = \dfrac{1}{1+\frac{1}{s_2}\sum\limits_{i=1}^{s_2}\frac{s_i^{Y_{IWT}}}{Y_{IWT_{i0}}}}$, Formula (3) can be linearized as Formula (4):

$$E_0^{IWT} = \min k - \frac{1}{p_1+m_2}\left(\sum_{i=1}^{p_1}\frac{s_i^{P_{IED}^{*}}}{P_{P_{IED_{i0}}}} + \sum_{i=1}^{m_2}\frac{s_i^{X_{IWT}}}{X_{IWT_{i0}}}\right)$$

$$s.t.\begin{cases} X_{IWT}\Lambda^{IWT} + S^{X_{IWT}} - kX_{IWT_0} = 0 \\ P_{IED}\Lambda^{IWT} + S^{P_{IED}^{*}} - kP_{IED_0} = 0 \\ Y_{IWT}\Lambda^{IWT} - S^{Y_{IWT}} - kY_{IWT_0} = 0 \\ k + \frac{1}{s_2}\sum\limits_{i=1}^{s_2}\frac{s_i^{Y_{IWT}}}{Y_{IWT_{i0}}} = 1 \\ \frac{1}{m_1}\left(\sum\limits_{i=1}^{m_1}\frac{S_i^{X_{IED}}}{X_{IED_{i0}}}\right) + \frac{E_0^{IED^{*}}}{s_1+p_1}\left(\sum\limits_{i=1}^{s_1}\frac{S_i^{Y_{IED}}}{Y_{IED_{i0}}} + \sum\limits_{i=1}^{p_1}\frac{S_i^{P_{IED}}}{P_{IED_{i0}}}\right) + k(E_0^{IED^{*}} - 1) = 0 \\ P_{IED}\lambda^{IWT} - P_{IED}\lambda^{IED} = 0 \\ S^{X_{IWT}} \geq 0, S^{P_{IED}^{*}} \geq 0, S^{Y_{IWT}} \geq 0, S^{X_{IED}} \geq 0, S^{Y_{IED}} \geq 0, S^{P_{IED}} \geq 0 \\ \Lambda^{IWT} \geq 0, \Lambda^{IED} \geq 0 \\ k \geq 0 \end{cases} \quad (4)$$

where $S^{X_{IED}} = ks^{X_{IED}}, S^{Y_{IED}} = ks^{Y_{IED}}, S^{P_{IED}} = ks^{P_{IED}}, S^{P_{IED}^{*}} = ks^{P_{IED}^{*}}, S^{X_{IWT}} = ks^{X_{IWT}}, S^{Y_{IWT}} = ks^{Y_{IWT}},$
$\Lambda^{IED} = k\lambda^{IED}$, and $\Lambda^{IWT} = k\lambda^{IWT}$.

Similarly, for IWT-dominated provinces with IED assistance, it is necessary to maximize the efficiency of the IWT stage first and then maximize the efficiency of the IED stage under the constraint that the IWT stage efficiency remains unchanged. The corresponding models are defined in Formulas (5) and (6):

$$\theta_0^{IWT} = \min\frac{1-\frac{1}{p_1+m_2}\left(\sum\limits_{i=1}^{p_1}\frac{s_i^{P_{IED}}}{P_{IED_{i0}}} + \sum\limits_{i=1}^{m_2}\frac{s_i^{X_{IWT}}}{X_{IWT_{i0}}}\right)}{1+\frac{1}{s_2}\sum\limits_{i=1}^{s_2}\frac{s_i^{Y_{IWT}}}{Y_{IWT_{i0}}}}$$

$$s.t.\begin{cases} X_{IWT}\lambda^{IWT} + s^{X_{IWT}} = X_{IWT_0} \\ P_{IED}\lambda^{IWT} + s^{P_{IED}} = P_{IED_0} \\ Y_{IWT}\lambda^{IWT} - s^{Y_{IWT}} = Y_{IWT_0} \\ s^{X_{IWT}} \geq 0, s^{P_{IED}} \geq 0, s^{Y_{IWT}} \geq 0 \\ \lambda^{IWT} \geq 0 \end{cases} \quad (5)$$

where $\lambda^{IWT}$ is the weight vector of the IWT stage, and $s^{X_{IWT}}, s^{P_{IED}}$, and $s^{Y_{IWT}}$ are the slack variables of $X_{IWT}, P_{IED}$, and $Y_{IWT}$, respectively.

$$\theta_0^{IED} = \min\frac{1-\frac{1}{m_1}\left(\sum\limits_{i=1}^{m_1}\frac{s_i^{X_{IED}}}{X_{IED_{i0}}}\right)}{1+\frac{1}{s_1+p_1}\left(\sum\limits_{i=1}^{s_1}\frac{s_i^{Y_{IED}}}{Y_{IED_{i0}}} + \sum\limits_{i=1}^{p_1}\frac{s_i^{P_{IED}^{*}}}{P_{IED_{i0}}}\right)}$$

$$s.t.\begin{cases} X_{IED}\lambda^{IED} + s^{X_{IED}} = X_{IED_0} \\ Y_{IED}\lambda^{IED} - s^{Y_{IED}} = Y_{IED_0} \\ P_{IED}\lambda^{IED} + s^{P_{IED}^{*}} = P_{IED_0} \\ \dfrac{1-\frac{1}{p_1+m_2}\left(\sum\limits_{i=1}^{p_1}\frac{s_i^{P_{IED}}}{P_{IED_{i0}}} + \sum\limits_{i=1}^{m_2}\frac{s_i^{X_{IWT}}}{X_{IWT_{i0}}}\right)}{1+\frac{1}{s_2}\sum\limits_{i=1}^{s_2}\frac{s_i^{Y_{IWT}}}{Y_{IWT_{i0}}}} = E_0^{IWT^{*}} \\ P_{IED}\lambda^{IWT} = P_{IED}\lambda^{IED} \\ s^{X_{IED}} \geq 0, s^{Y_{IED}} \geq 0, s^{P_{IED}^{*}} \geq 0, s^{X_{IWT}} \geq 0, s^{P_{IED}} \geq 0, s^{Y_{IWT}} \geq 0 \\ \lambda^{IED} \geq 0, \lambda^{IWT} \geq 0 \end{cases} \quad (6)$$

In Formula (8), $\lambda^{IED}$ and $\lambda^{IET}$ are the weight vectors of the IED and IWT stages, respectively.

According to Hu et al. [6], the total-factor water efficiency of the IED stage ($\phi^{IED}$) can be described as follows:

$$\phi^{IED} = \frac{Actual\ water\ consumption - slack\ of\ water\ input}{Actual\ water\ consumption}. \tag{7}$$

Further, the total-factor water efficiency of the IWT stage ($\phi^{IWT}$) can be described as follows:

$$\phi^{IWT} = \frac{Actual\ wastewater\ treatment}{Actual\ wastewater\ treatment + slack\ of\ wastewater\ treatment}. \tag{8}$$

Based on existing studies [32,33], we defined the overall efficiency of $DMU_0$ for two types of regions, as shown in Formula (9):

$$\phi = \sqrt{\phi^{IED} \times \phi^{IWT}} \tag{9}$$

(1) If $\phi^{IED}(\phi^{IWT}) = 1$, the slack variables are equal to 0 in a given stage, and $DMU_0$ in the IED (or IWT) stage is efficient.
(2) If $\phi^{IED}(\phi^{IWT}) \prec 1$, $DMU_0$ in the IED (or IWT) stage is inefficient.
(3) If and only if $\phi^{IED} = 1, \phi^{IWT} = 1$ and the slack variables are 0 in each stage, then $DMU_0$ is efficient.

### 2.2. Regression Analysis of Determinants

To explore the effect of urbanization on industrial water consumption efficiency in China, a multiple linear regression model should be constructed. Because industrial water consumption efficiency values are within the range of $[0,1]$, a common practice for analysis of determinants is using the Tobit estimator. However, as demonstrated by Simar and Wilson [34], this is inappropriate. Because industrial water consumption efficiency is not observed but estimated by DEA, it is difficult to assume that error terms distribute independently. In addition, industrial water consumption efficiency is estimated based on the sample of provinces, so the estimate of efficiency is biased. On this basis, the Simar–Wilson procedure was proposed, which was based on a double bootstrap that enables consistent inference within models explaining efficiency scores while simultaneously producing standard errors and confidence intervals for these efficiency scores. This study treated industrial water consumption efficiency as the explained variable and assumed and tested the following regression specification:

$$y_i = \alpha + x_i\beta + \mu_i \tag{10}$$

In Formula (10), $y_i$ is the explained variable, $\beta$ is the regression coefficient, $x_i$ is the explained variable vector, and $\mu_i$ is statistical noise.

### 2.3. Variables and Data

(1) The inputs and outputs of the industrial water consumption efficiency model

This paper referred to some existing studies [11,17,27] and considered industrial development investment, industrial employment population, industrial water consumption, and investment in IWT as inputs. Additionally, the industrial economic output (GDP), actual wastewater discharge amount, and water savings amount were the model outputs. The indicators are shown in Table 1.

**Table 1.** Indicators used for industrial water consumption evaluation.

| Indicator Types | Indicators | Variables and Units |
|---|---|---|
| $X_{IED}$ | Industrial Investment | Industrial Development Investment ($10^8$ yuan) |
| | Industrial Labor | Industrial Employment Population ($10^4$) |
| | Industrial Water Consumption | Industrial Water Consumption ($10^4$ m$^3$) |
| $P_{IED}$ | Wastewater Discharge in the IED Stage | Initial Industrial Wastewater Discharge Amount ($10^4$ m$^3$) |
| $Y_{IED}$ | Industrial Output | Industrial Economic Output ($10^8$ yuan) |
| $X_{IWT}$ | Government Investment | Wastewater Treatment Investment ($10^8$ yuan) |
| $Y_{IWT}$ | Wastewater Treatment | Industrial Wastewater Treatment Amount ($10^4$ m$^3$) |

The data in Table 1 were collected from the China Statistical Yearbook (2011–2016) [1] and China Environmental Statistics Yearbook (2011–2016) [35]. Some data for Hong Kong, Macao, Taiwan, and Tibet were missing, so data from 30 provinces in China collected between 2011 and 2015 were used in this study. The data on industrial investment, industrial output, and government investment were converted into constant price in 2011.

(2)    The relevant influential factors

Industrial water consumption efficiency is affected by many factors. Specifically, with the continuous development of urbanization in China, industrial water consumption efficiency is affected as follows.

(1) Urbanization promotes regional economic development, which can be characterized by the corresponding per capita income. With the increase in employee income, support for environmental protection will increase, especially for water protection. As a result, the local government and enterprises will increase investments in IWT and industrial upgrades, which will affect the efficiency of industrial water consumption. Therefore, we chose the per capita disposable income of urban residents (*pdi*) as an explanatory variable to reflect the effect of urbanization on regional economic development.

(2) With the development of urbanization, the regional industrial structure changes accordingly, especially for secondary industries, which rapidly develop. As a result, the demand for industrial water will increase [36,37], and the pressure for wastewater treatment will increase. These factors affect the efficiency of industrial water consumption. Therefore, we chose the industrial development percentage (*ins*) and industrial water consumption percentage (*iwp*) as explanatory variables to reflect the effect of urbanization on the regional industrial structure.

(3) When urbanization occurs, people move from rural to urban areas. On one hand, the population increase in urban areas provides a large labor force, which helps promote industrial outputs [23]. On the other hand, the population increase in urban areas will result in the increased consumption of domestic water [38], which will reduce the inputs for industrial water consumption. Therefore, we chose the urban population density (*upd*) and urban population (*urp*) as explanatory variables to reflect the effect of urbanization on regional population growth.

(4) The urban area will expand continuously during urbanization. Buildings and asphalt roads will cover an increasing percentage of the land, which will decrease rainwater infiltration and lead to more evaporation of rainwater. Therefore, we chose the percentages of built-up areas in urban areas (*pba*) as explanatory variables to reflect the effect of regional spatial changes caused by urbanization.

To control the effects of other variables, we used the R&D funds of industrial enterprises (*RD*) and foreign investment (*foi*) as control variables based on the studies of Han et al., Fan et al., Chen et al., Long et al., and Deng et al. [9,16,39–42]. The indicators are shown in Table 2.

**Table 2.** Industrial water consumption efficiency indicators.

| Influential Factors | Variable Names | Variable Meanings |
|---|---|---|
| Economic development | *pdi* | Per capita disposable income of urban residents ($10^4$ yuan) |
| Industrial structure | *ins* | Industrial development proportion (%) |
| | *iwp* | Industrial water consumption proportion (%) |
| Population growth | *upd* | Urban population density (100 people per square kilometer) |
| | *urp* | Urban population ($10^6$) |
| Spatial change | *pba* | Proportion of built-up areas (%) |
| Control variables | *RD* | R&D funds of industrial enterprises ($10^8$ yuan) |
| | *foi* | Foreign investment ($10^9$ yuan) |

All the data for the abovementioned indicators were collected from the *China Statistical Yearbook (2011–2016)* and *China Environmental Statistics Yearbook (2011–2016)*. The data on GDP, R&D, and foreign investment were converted into constant price in 2011.

## 3. Results and Discussion

### 3.1. Measurement of the Efficiency of Industrial Water Consumption

Based on the results of some previous studies [21,22,27], 30 provinces (excluding Hong Kong, Macao, Taiwan, and Tibet) in China were divided into IED-dominated and IWT-dominated provinces, as shown in Table 3.

**Table 3.** Regional classification based on the master–slave relationship.

| Categories | Provinces |
|---|---|
| IED-dominated regions | Inner Mongolia, Jilin, Henan, Xinjiang, Gansu, Ningxia, Qinghai, Sichuan, Yunnan, Anhui, Hubei, Jiangxi, Hunan, Guizhou, Guangxi, Hainan |
| IWT-dominated regions | Liaoning, Beijing, Tianjin, Hebei, Shandong, Shanxi, Shaanxi, Heilongjiang, Chongqing, Jiangsu, Shanghai, Zhejiang, Fujian, Guangdong |

Tables 4 and 5 report the two-stage efficiency results for the 30 provinces using the proposed model.

Overall, there was no significant increase in the efficiency of industrial water consumption in the two groups of regions from 2011 to 2015. The efficiency of each stage in both regions was significantly different from the perspective of coefficient of variation (*CV*). The efficiency in the IED stage was higher than that in the IWT stage in IWT-dominated regions, while the opposite situation occurred in IED-dominated regions. This indicated that most of the provinces focused more on industrial economic development and less on industrial wastewater treatment in IWT-dominated regions. For IED-dominated regions, it is important to intensively use water resources in industry production. In addition, it must be pointed that the efficiency of the IWT stage of IED-dominated regions is higher than the IED stage because of the slow development of industry. Therefore, it is very important to develop low-pollution industries to simultaneously promote the efficiency of the IED and IWT stages.

As shown in Figure 2, in IED-dominated regions, the average efficiencies in two stages and the overall efficiency increased to some extent from 2011 to 2014 and decreased in 2015. In IWT-dominated regions, the average efficiency in the IED stage had no significant change from 2011 to 2015, and the efficiency of each year was close to the DEA effective level. The average efficiency in the IWT stage showed a visible fluctuation from 2011 to 2015, and there was no significant increase from a holistic perspective. By contrast, the IED stage efficiency in IWT-dominated regions was greater than that in IED-dominated regions from 2011 to 2015 because the IWT-dominated regions achieved rapid

industrial development and industry slowly developed in the IED-dominated regions. Therefore, wastewater treatment, as a type of "feedback" associated with industrial output, is necessary in IWT-dominated regions. Moreover, promoting the coordinated development of the economy and the environment is necessary in IED-dominated regions.

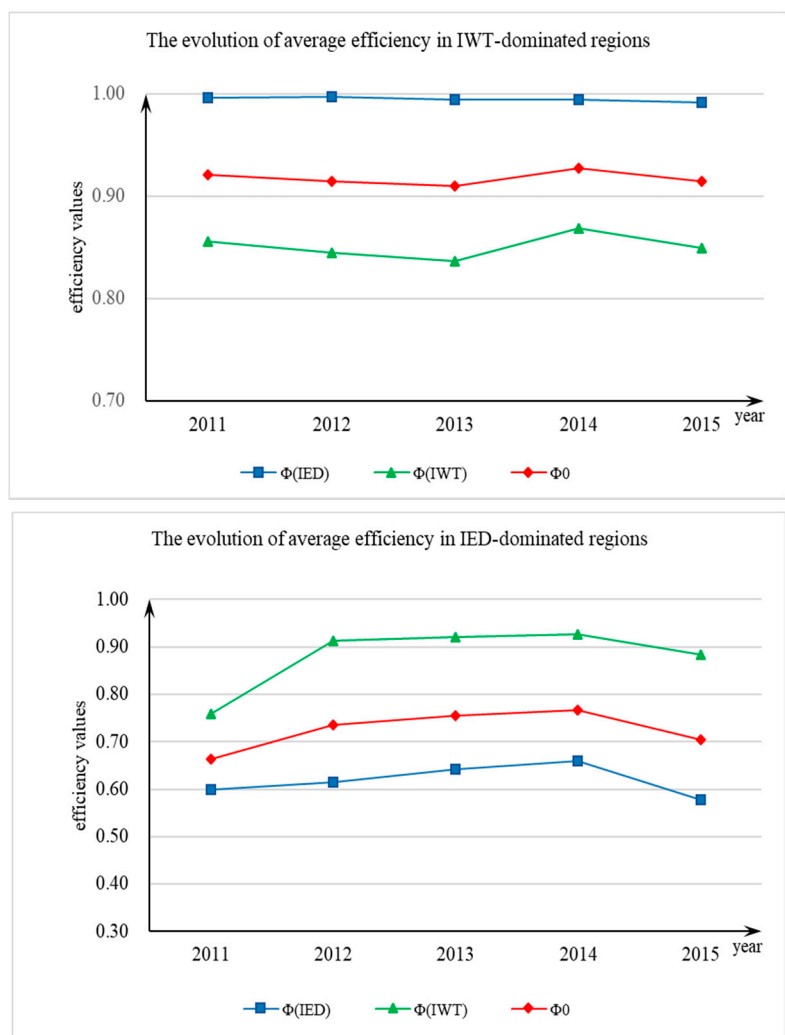

**Figure 2.** The evolution of the average efficiency in two types of regions.

**Table 4.** Efficiency of industrial water consumption in industrial economic development (IED)-dominated regions.

| IED-Dominated Regions | 2011 | | | 2012 | | | 2013 | | | 2014 | | | 2015 | | |
|---|---|---|---|---|---|---|---|---|---|---|---|---|---|---|---|
| | $\phi_0^{IED}$ | $\phi_0^{IWT}$ | $\phi_0$ | $\phi_0^{IED}$ | $\phi_0^{IWT}$ | $\phi_0$ | $\phi_0^{IED}$ | $\phi_0^{IWT}$ | $\phi_0$ | $\phi_0^{IED}$ | $\phi_0^{IWT}$ | $\phi_0$ | $\phi_0^{IED}$ | $\phi_0^{IWT}$ | $\phi_0$ |
| Inner Mongolia | 1.000 | 0.738 | 0.859 | 1.000 | 0.988 | 0.994 | 1.000 | 1.000 | 1.000 | 1.000 | 0.818 | 0.905 | 1.000 | 0.828 | 0.910 |
| Jilin | 0.637 | 0.707 | 0.671 | 0.680 | 0.910 | 0.787 | 1.000 | 0.774 | 0.880 | 1.000 | 1.000 | 1.000 | 0.646 | 0.767 | 0.704 |
| Anhui | 0.414 | 0.778 | 0.567 | 0.390 | 1.000 | 0.625 | 0.395 | 1.000 | 0.628 | 0.399 | 1.000 | 0.632 | 0.398 | 0.985 | 0.626 |
| Jiangxi | 0.440 | 0.716 | 0.561 | 0.458 | 1.000 | 0.677 | 0.458 | 0.908 | 0.645 | 0.459 | 1.000 | 0.678 | 0.445 | 0.893 | 0.631 |
| Henan | 1.000 | 0.707 | 0.841 | 1.000 | 0.780 | 0.883 | 0.935 | 0.766 | 0.846 | 1.000 | 0.767 | 0.876 | 0.770 | 0.778 | 0.774 |
| Hubei | 0.418 | 0.786 | 0.573 | 0.403 | 1.000 | 0.635 | 0.417 | 1.000 | 0.646 | 0.499 | 1.000 | 0.706 | 0.437 | 1.000 | 0.661 |
| Hunan | 0.439 | 0.740 | 0.570 | 0.440 | 1.000 | 0.664 | 0.489 | 0.998 | 0.699 | 0.554 | 1.000 | 0.744 | 0.448 | 1.000 | 0.669 |
| Guangxi | 0.486 | 0.818 | 0.630 | 0.507 | 1.000 | 0.712 | 0.484 | 1.000 | 0.696 | 0.459 | 0.993 | 0.675 | 0.452 | 1.000 | 0.672 |
| Hainan | 0.512 | 0.707 | 0.602 | 0.519 | 0.748 | 0.623 | 0.464 | 1.000 | 0.682 | 0.473 | 1.000 | 0.688 | 0.495 | 0.762 | 0.614 |
| Sichuan | 0.592 | 0.827 | 0.700 | 0.646 | 0.954 | 0.785 | 0.711 | 0.949 | 0.821 | 0.948 | 0.957 | 0.953 | 0.554 | 0.838 | 0.681 |
| Guizhou | 0.409 | 1.000 | 0.639 | 0.376 | 1.000 | 0.613 | 0.468 | 1.000 | 0.684 | 0.464 | 1.000 | 0.681 | 0.473 | 1.000 | 0.687 |
| Yunnan | 0.552 | 0.783 | 0.657 | 0.535 | 1.000 | 0.731 | 0.587 | 1.000 | 0.766 | 0.550 | 1.000 | 0.742 | 0.557 | 0.983 | 0.740 |
| Gansu | 0.451 | 0.710 | 0.566 | 0.478 | 0.776 | 0.609 | 0.545 | 0.780 | 0.652 | 0.496 | 0.755 | 0.612 | 0.470 | 0.760 | 0.597 |
| Qinghai | 0.803 | 0.716 | 0.758 | 1.000 | 0.911 | 0.955 | 0.937 | 1.000 | 0.968 | 0.983 | 1.000 | 0.992 | 0.793 | 1.000 | 0.890 |
| Ningxia | 0.687 | 0.707 | 0.697 | 0.661 | 0.762 | 0.710 | 0.657 | 0.780 | 0.716 | 0.609 | 0.756 | 0.678 | 0.655 | 0.765 | 0.708 |
| Xinjiang | 0.760 | 0.716 | 0.738 | 0.741 | 0.768 | 0.755 | 0.731 | 0.793 | 0.761 | 0.660 | 0.768 | 0.712 | 0.647 | 0.795 | 0.717 |
| Average | 0.600 | 0.760 | 0.664 | 0.615 | 0.912 | 0.735 | 0.642 | 0.922 | 0.756 | 0.660 | 0.926 | 0.767 | 0.577 | 0.885 | 0.705 |
| CV | 0.332 | 0.1 | 0.145 | 0.357 | 0.116 | 0.164 | 0.339 | 0.112 | 0.154 | 0.358 | 0.117 | 0.171 | 0.286 | 0.12 | 0.126 |

**Table 5.** Efficiency of industrial water consumption in industrial wastewater treatment (IWT)-dominated regions.

| IWT-Dominated Regions | 2011 | | | 2012 | | | 2013 | | | 2014 | | | 2015 | | |
|---|---|---|---|---|---|---|---|---|---|---|---|---|---|---|---|
| | $\phi_0^{IED}$ | $\phi_0^{IWT}$ | $\phi_0$ | $\phi_0^{IED}$ | $\phi_0^{IWT}$ | $\phi_0$ | $\phi_0^{IED}$ | $\phi_0^{IWT}$ | $\phi_0$ | $\phi_0^{IED}$ | $\phi_0^{IWT}$ | $\phi_0$ | $\phi_0^{IED}$ | $\phi_0^{IWT}$ | $\phi_0$ |
| Beijing | 0.972 | 0.885 | 0.927 | 0.964 | 0.973 | 0.969 | 0.967 | 0.843 | 0.903 | 0.979 | 0.962 | 0.970 | 0.908 | 0.969 | 0.938 |
| Tianjin | 0.991 | 0.947 | 0.969 | 0.998 | 0.957 | 0.977 | 0.963 | 0.934 | 0.948 | 0.954 | 0.895 | 0.924 | 0.985 | 0.963 | 0.974 |
| Hebei | 1.000 | 1.000 | 1.000 | 1.000 | 1.000 | 1.000 | 1.000 | 1.000 | 1.000 | 1.000 | 1.000 | 1.000 | 1.000 | 1.000 | 1.000 |
| Shanxi | 1.000 | 0.978 | 0.989 | 1.000 | 0.914 | 0.956 | 1.000 | 0.697 | 0.835 | 1.000 | 0.955 | 0.977 | 1.000 | 0.693 | 0.832 |
| Liaoning | 1.000 | 0.897 | 0.947 | 1.000 | 0.955 | 0.977 | 1.000 | 0.803 | 0.896 | 1.000 | 1.000 | 1.000 | 1.000 | 0.969 | 0.984 |
| Heilongjiang | 1.000 | 0.860 | 0.927 | 1.000 | 0.933 | 0.966 | 1.000 | 0.879 | 0.937 | 1.000 | 1.000 | 1.000 | 1.000 | 0.726 | 0.852 |
| Shanghai | 1.000 | 1.000 | 1.000 | 1.000 | 0.996 | 0.998 | 0.994 | 0.813 | 0.899 | 0.996 | 0.684 | 0.826 | 0.985 | 0.720 | 0.842 |
| Jiangsu | 0.995 | 0.753 | 0.866 | 1.000 | 0.708 | 0.842 | 1.000 | 0.896 | 0.947 | 1.000 | 0.857 | 0.925 | 1.000 | 0.688 | 0.829 |
| Zhejiang | 1.000 | 0.682 | 0.826 | 1.000 | 0.715 | 0.845 | 1.000 | 0.681 | 0.825 | 1.000 | 0.695 | 0.834 | 1.000 | 0.687 | 0.829 |
| Fujian | 0.998 | 0.678 | 0.822 | 1.000 | 0.699 | 0.836 | 1.000 | 0.995 | 0.998 | 1.000 | 0.751 | 0.867 | 1.000 | 0.830 | 0.911 |
| Shandong | 0.999 | 0.715 | 0.845 | 1.000 | 0.695 | 0.834 | 1.000 | 0.694 | 0.833 | 1.000 | 0.725 | 0.851 | 1.000 | 0.854 | 0.924 |
| Guangdong | 0.999 | 0.973 | 0.986 | 0.995 | 0.760 | 0.869 | 0.993 | 0.768 | 0.873 | 0.997 | 0.807 | 0.897 | 0.999 | 0.902 | 0.949 |
| Chongqing | 0.997 | 0.948 | 0.972 | 1.000 | 0.831 | 0.912 | 1.000 | 0.707 | 0.841 | 1.000 | 1.000 | 1.000 | 1.000 | 0.954 | 0.976 |
| Shaanxi | 1.000 | 0.672 | 0.820 | 1.000 | 0.688 | 0.830 | 1.000 | 0.997 | 0.999 | 1.000 | 0.824 | 0.908 | 1.000 | 0.936 | 0.968 |
| Average | 0.996 | 0.856 | 0.921 | 0.997 | 0.845 | 0.915 | 0.994 | 0.836 | 0.910 | 0.995 | 0.868 | 0.927 | 0.991 | 0.849 | 0.915 |
| CV | 0.008 | 0.15 | 0.077 | 0.01 | 0.151 | 0.075 | 0.013 | 0.141 | 0.07 | 0.013 | 0.139 | 0.07 | 0.025 | 0.144 | 0.071 |

### 3.2. Analysis of the Influential Factors

According to Formula (10) and Table 2, in order to analyze the determinants of industrial water efficiency, we adopted the approach of Simar and Wilson [34]. The regression model was established as shown in Formula (11):

$$Y_{it} = \beta_0 + \beta_1 \times pdi_{it} + \beta_2 \times ins_{it} + \beta_3 \times upd_{it} + \beta_4 \times urp_{it} + \beta_5 \times pba_{it} + \beta_6 \times foi_{it} + \beta_7 \times RD_{it} + \beta_8 \times iwp_{it} + \varepsilon_{it} \tag{11}$$

where $Y_{it}$ represents the efficiency of province $i$ in year $t$, $\beta_j (j = 0, 1, 2, \ldots, 9)$ represents the regression coefficients, and $\varepsilon_{it}$ represents the statistical noise. The results are shown in Table 6.

**Table 6.** The regression-based industrial water consumption efficiency results.

| Classifications | Indicators | IED-Dominated Regions | | | IWT-Dominated Regions | | |
|---|---|---|---|---|---|---|---|
| | | $\phi_0^{IED}$ | $\phi_0^{IWT}$ | $\phi_0$ | $\phi_0^{IED}$ | $\phi_0^{IWT}$ | $\phi_0$ |
| Economic development | *pdi* | −0.209 | −0.344 (***) | −0.088 (**) | 0.013 (***) | 0.159 (**) | 0.062 (**) |
| Industrial structure | *ins* | 0.03 (***) | 0.004 | 0.017 (*) | 0.001 (*) | −0.006 | −0.001 |
| | *iwp* | −0.052 (***) | 0.018 (***) | −0.015 (**) | −0.008 (**) | 0.013 | 0.004 |
| Population growth | *upd* | 0.045 | 0.015 | 0.024 | 0.009 (***) | 0.013 (*) | 0.005 (*) |
| | *urp* | 0.010 | −0.007 | 0.002 | −0.011 (***) | 0.010 (**) | 0.003 (***) |
| Spatial change | *pba* | −0.039 | −0.01 (***) | −0.010 (***) | 0.007 (*) | 0.018 (**) | 0.071 (*) |
| Control variables | *R&D* | 0.010 (*) | 0.011 (***) | 0.003 (***) | 0.002 (***) | 0.002 (**) | 0.003 (***) |
| | *foi* | 0.016 (***) | −0.015 (*) | −0.011 (***) | −0.005 (**) | 0.008 (**) | 0.003 |
| *Cons.* | | 2.547 (*) | 2.169 (**) | 2.532 (*) | 1.147 | −0.288 (*) | 0.638 (**) |

Note: *, **, *** represent passing significant levels of 10%, 5% and 1%, respectively.

According to Table 6, the influential factors have different effects on the efficiency of industrial water consumption, and the same factor can have significantly different effects in different regions.

(1)  Analysis of the effect of economic development

Notably, the increase in the disposable income of urban residents (*pdi*) was negatively related to the efficiency of the IWT stage in IED-dominated regions (significant at the 1% level) and positively related to the efficiency of the IED stage in IWT-dominated regions (significant at the 1% level). This finding indicates that with the development of urbanization, the rapid growth in the disposable income of urban residents in IED-dominated regions results in an increased water demand, which leads to increases in industrial water consumption and industrial wastewater production. Moreover, the local governments of IED-dominated regions focus too much on economic development, resulting in an inefficient IWT stage.

IWT-dominated regions are industrially developed regions with advanced economic development. Therefore, people will move to these areas with the development of urbanization, and these urban residents will consume more water as the urban population increases, thereby decreasing the amount of

water available for industrial use. Moreover, with the growth of disposable income, the desire for clean water will increase, which may require investments in industrial production and wastewater treatment.

(2)   Analysis of the effect of the industrial structure

In the two types of regions, the secondary industry development (*ins*) was positively related to the efficiency of the IED stage and displayed no significant correlation with the IWT stage efficiency, suggesting that industrial development quickly promotes GDP growth and increases the efficiency of the IED stage but cannot significantly increase the efficiency of the IWT stage. However, it is important to point out that the rapid development of tertiary industries will consume a large amount of water and produce a large amount of wastewater, which increases the burden associated with industrial wastewater treatment, especially in IWT-dominated regions. Similar to the trend observed for secondary industry development (*ins*), industrial water consumption (*iwp*) was positively related to the efficiency of the IED stage and had no significant correlation with the IWT stage efficiency in both types of regions.

(3)   Analysis of the effect of population growth

The urban population density (*upd*) had significantly different effects on consumption patterns in different types of regions. In IED-dominated regions, the effect of the urban population density was not significant, and in IWT-dominated regions, the urban population density was positively related to the efficiency of each stage and the overall efficiency. This finding indicates that in IWT-dominated regions, the agglomerated population provides many competent laborers for industrial production and wastewater treatment, and in IED-dominated regions, the effect of population agglomeration is not apparent. Based on the urban population (*urp*) results, we see that population growth has the same effect as the population density. However, if too many people relocate to cities, the increased water demand will increase the wastewater treatment requirements, especially in IWT-dominated regions.

(4)   Analysis of the effect of spatial changes

In IED-dominated regions, the proportion of the built-up area (*pba*) displayed a significant negative correlation with the efficiency of the IWT stage and the overall efficiency, and this correlation was significant at the 1% level. No significant correlation was observed with the IED stage efficiency. This result may be caused by incomplete infrastructure construction and imperfect industrial layouts during spatial expansion in the process of urbanization. In IWT-dominated regions, the proportion of the urban area exhibits a significant positive correlation with the efficiency of each stage and the overall efficiency. Thus, the urban space in IWT regions reaches full capacity. An increase in the urban area provides more space for industrial production and wastewater treatment, leading to improved efficiency in each stage in IWT-dominated regions. Moreover, the expansion of urban areas is conducive to industries that can adjust their economic strategies based on local conditions, thereby promoting economic development and industrial upgrades.

(5)   Analysis of the effect of control variables

*R&D* was positively related to the efficiency of each stage in both regions, indicating that *R&D* can provide advanced production and wastewater treatment equipment and technology to potentially change the industrial production mode. Such a change may increase the industrial output and decrease the consumption of water and the discharge of wastewater. Also, such a change may increase the wastewater treatment capacity and promote the efficiency of the IWT stage of both regions.

In IED-dominated regions, foreign investment (*foi*) displayed a significant negative correlation with the IWT stage efficiency and overall efficiency and a significant positive correlation with the IED stage efficiency. In IWT-dominated regions, foreign investment was negatively correlated with the IED stage efficiency. Therefore, foreign investment can improve the industrial output in IED-dominated regions, but a large amount of wastewater is often produced, which is a notable burden for the efficiency of the IWT stage.

### 4. Conclusions and Policy Implications

This study established a DEA model based on master–slave game relationships and measured the two-stage and overall efficiency of industrial water consumption in IED- and IWT-dominated regions in China. The effects of urbanization on the efficiency of each stage and the overall efficiency were further analyzed using a regression model. The conclusions of the study are as follows: (1) There was no significant increase in the efficiency of industrial water consumption in the two types of regions from 2011 to 2015. The efficiency of the IED stage was consistently larger than that of the IWT stage in IWT-dominated regions. Moreover, the IED stage efficiency in IWT-dominated regions was always larger than that in IED-dominated regions from 2011 to 2015. (2) Urbanization affects industrial water consumption efficiency from multiple perspectives. Moreover, the effects of the same factor can differ in different regions, and the effects of the same factor can differ in different stages in the same region.

Based on the results of this study, some policy suggestions are as follows.

(1)    For IED-dominated regions

First, with the development of urbanization, more people will move to urban areas. Local governments should moderately increase the population density to supply competent laborers for industrial production and wastewater treatment. Second, local governments should promote reasonable consumption and green lifestyles among residents to save water. Third, to improve the economy, it is necessary to encourage an industry shift in IWT-dominated regions. Additionally, it is important for IED-dominated regions to develop water-saving industries. Fourth, although the growth of built-up and urban areas has no significant effect on industrial water consumption, local governments should rationally optimize the industrial and urban spatial layouts. Specifically, the wastewater treatment infrastructure should match the industrial development pattern.

It can be seen that the overall efficiency of industrial water consumption of IED-dominated regions was lower. Particularly, the overall efficiency values of industrial water consumption of Anhui, Jiangxi, Hubei, Hunan, and Gansu were lower than 0.6, and both of the efficiencies of the IED and IWT stages in these provinces were very low. So, it is important to formulate a clear industry development strategy to promote the efficiency of IED and IWT stages simultaneously. Also, it may be more helpful to achieve sustainable industrial development by undertaking low-pollution industry transfers according to local resource endowments. Meanwhile, learning from the experience of developed regions and taking environmental measures in advance are necessary for undertaking industry transfer. Furthermore, local governments should facilitate the importation of advanced water-saving production and wastewater treatment technologies for local industry enterprises.

(2)    For IWT-dominated regions

The population density in IWT-dominated regions has reached a peak. Although a higher population density can provide many laborers for industrial production and wastewater treatment, many IWT-dominated regions are shifting toward energy-intensive and water-intensive industries and developing productive service industries. Therefore, rationally promoting relocation from core cities to satellite cities will be conducive to decreasing residential water consumption and the wastewater treatment load. Additionally, the backward production capacity in various industries should be eliminated. On one hand, most of the backward production capacity is associated with water-intensive industries that consume a large amount of water and energy in the production process and produce considerable wastewater. Therefore, eliminating backward production capacity can save water. On the other hand, most of backward production capacity has been prevalent for a long time, and the corresponding industries occupy large proportions of urban land. Therefore, eliminating backward production capacity can provide some space for new industries. Finally, the land in many IWT regions is largely occupied by buildings and asphalt roads, and rainwater infiltration is limited. In such cases, most of the rainwater evaporates or forms runoff. Therefore, decreasing the proportion of built-up

areas and increasing urban green areas can improve rainwater storage and promote groundwater system recovery.

(3)  For both types of regions

First, economic development can provide advanced production equipment and technology, as well as advanced wastewater treatment technology. Therefore, local governments in IED- and IWT-dominated regions should promote economic development to achieve industrial upgrades. Second, the promotion of science and technology can improve the efficiency of industrial water consumption. Therefore, local governments should encourage research institutions to study the core problems associated with industrial production processes and wastewater treatment to improve the efficiency of industrial water consumption. Additionally, it is important to attract talented researchers and laborers in urban areas. Third, although foreign investment can improve industrial output, environmental pollution must be taken into account when introducing foreign investment.

**Author Contributions:** This paper was written based on Master Thesis of Y.L. Study process of the paper was conducted mainly by B.L. R.H. has adjusted the format of the paper to fit the style of Sustainability. H.W. has reinforced the references to meet the reviewer's requirements. B.L. and R.H. have prepared the explanation to reviewers in Response Letter.

**Funding:** This research was funded by National Natural Science Foundation of P.R. China (No.71804189, NO. 71501188), the Shandong Provincial Natural Science Foundation of P.R. China (NO: ZR2016GQ06), the Ministry of Education of Humanities and Social Science project (NO: 16YJAZH054, 18YJC630176), and the Fundamental Research Funds for the Central Universities (NO: 17CX04017B).

**Conflicts of Interest:** No potential conflict of interests is reported by authors.

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
