# Peer review of "Does Urbanization Improve Industrial Water Consumption Efficiency?"

_sustainability, doi:10.3390/su11061787_

Round 1

Reviewer 1 Report

overall the hypehesis to be examned is of high nterest to policy makers and the academia.

However there are issues need to be improved , see for example paper ï‚™ Zervopoulos, P. amnd  Palaskas, T. (2011) ‘Applying Quality-Driven, Efficiency-Adjusted DEA (QE-DEA) in the Pursuit of High-Efficiency – High Quality Service Units: An Input-Oriented Approach’, IMA Journal of Management Mathematics 22 (4), 401-417.

Is not sufficient to say the efficiency differs. Do you mean statistically?

Author Response

Thank you for your detailed comments on our manuscript. we have revised our manuscript carefully. the response is as followed, and a pdf file is uploaded in the attachment.thank you!

Point 1: However there are issues need to be improved , see for example paper ™ Zervopoulos, P. amnd Palaskas, T. (2011) ‘Applying Quality-Driven, Efficiency-Adjusted DEA (QE-DEA) in the Pursuit of High-Efficiency – High Quality Service Units: An Input-Oriented Approach’, IMA Journal of Management Mathematics 22 (4), 401-417. 

Response 1: Thank you for the reviewer’s detailed comment on this issue. We searched for the reference that you suggest. Unfortunately, we cannot download the reference. But we studied the reference from internet, and reorganized the manuscript according to the reference. Thank you!

Point 2: Is not sufficient to say the efficiency differs. Do you mean statistically?

Response 2: Thank you for the reviewer’s detailed comment on this issue. In order to explain the difference of efficiency of IED-stage, IWT-stage and overall efficiency, we using coefficient of variation (CV) to show the rule of the efficiency. Please check it in the table 4 and table 5.Thank you!

Some other revision.

1. We are aware of our limitation in English language, so after we complete the technical revision, we send our manuscript to professional English editing team, and pay for English editing.

2. We reread the manuscript thoroughly, and revised some spelling and language expression, in order to make it more readable.

Thank you for your suggestion for our manuscript.

Reviewer 2 Report

1.     The water consumption efficiency used in this paper is actual the overall efficiency of water and other inputs (p. 3), instead of the water consumption efficiency alone as a disaggregate input efficiency.  It is suggested that the water efficiency proposed by Hu et al. (2006) with target water input/actual water input should be used.  Another way to find the water efficiency is to treat the other inputs as quasi-fixed inputs such that water is the only input to adjust (reduce) in the DEA model.

2.     The DEA model used in this paper is a Tone’s slack-based measure (SBM) model.  It is not clear if this two-stage model is a recent network SBM model by Tone and Tsutsui.  If it is a network SBM by Tone and Tsutsui, their recent papers should be cited and briefly introduced.  If not, then further discussion on the network DEA model used in this paper should be added.

3.     Since this paper uses the panel data, the nominal variables should be deflated into real variables to remove the price level distortions.  However, in Table 2 on page 10, it is not clear whether or not the nominal variables have been deflated.  Otherwise, the efficiency scores may be distorted by changes in price levels.

4.     A Tobit regression on efficiency scores has been heavily criticized by Simar and Wilson for producing seriously biased estimates.  They proposed the double bootstrapping truncated regression to replace the Tobit regression.  Nowadays decent academic journals do not accept Tobit regressions on efficiency scores.  It is strongly suggested that the approach of Simar and Wilson should be followed.  Or the authors may use sign-rank tests to avoid the biased estimations by Tobit regressions.

5.     The empirical results by Tobit regression are somewhat inconsistent and difficult to explain.  For instance, R&D is efficiency-enhancing in one production stage but not in the other.  R&D is efficiency-enhancing is some provinces but not in others.  It is suggested that direct policy instruments on water consumption efficiency such as fees should be included and empirically tested.

6.     Many references on pp. 19-21 do not completely show all authors.  Some titles of papers are even wrong and do not match the exact full titles of the published articles.  They wrong titles may be translated into Chinese and then re-translated into English again.  Careful checking and proofreading are necessary before re-submission.

Author Response

Thank you for the your detailed comments. we have revised our manuscript careffully, and the response is as bellow. also, we upload a pdf file in the attachment. thank you!

Point 1: The water consumption efficiency used in this paper is actual the overall efficiency of water and other inputs (p. 3), instead of the water consumption efficiency alone as a disaggregate input efficiency. It is suggested that the water efficiency proposed by Hu et al. (2006) with target water input/actual water input should be used. Another way to find the water efficiency is to treat the other inputs as quasi-fixed inputs such that water is the only input to adjust (reduce) in the DEA model. 

Response 1: Thank you for the reviewer’s detailed comment on this issue. We are aware that the definition of industrial water consumption efficiency in our manuscript is not clear and not suitable. We adopt your suggestion, and redefine the efficiency of IED-stage, the efficiency of IWT-stage and the overall efficiency according to Hu et al. (2006). Please check it from line 152 to 160 on page 7. Thank you!

Point 2: The DEA model used in this paper is a Tone’s slack-based measure (SBM) model. It is not clear if this two-stage model is a recent network SBM model by Tone and Tsutsui. If it is a network SBM by Tone and Tsutsui, their recent papers should be cited and briefly introduced. If not, then further discussion on the network DEA model used in this paper should be added.

Response 2: Thank you for the reviewer’s detailed comment on this issue. The DEA model in our manuscript is a Tone’s slack-based measure (SBM) model, but it is not the recent network SBM model by Tone and Tsutsui. Indeed, the model is built according to Wang et al., 2015. And the overall efficiency is built based on existing studies (Zha et al., 2010; Yu et al., 2014). And also, we further discussed the relationship about the efficiency of each stage and the overall efficiency according to your suggestion. Please check it in the line 161 to 165 on page 7 and 8. Thank you!

Point 3: Since this paper uses the panel data, the nominal variables should be deflated into real variables to remove the price level distortions. However, in Table 2 on page 10, it is not clear whether or not the nominal variables have been deflated. Otherwise, the efficiency scores may be distorted by changes in price levels.

Response 3: Thank you for the reviewer’s detailed comment on this issue. We are aware that we have forgotten to report whether or not the nominal variables have been deflated. Indeed, the data of industrial investment, industrial output, government investment GDP, R&D and foreign investment are converted into constant price in 2011. Now we add it into the revised manuscript. Please check it in the line 187 on page 9, and in the line 230 on page 10. Thank you again!

Point 4: A Tobit regression on efficiency scores has been heavily criticized by Simar and Wilson for producing seriously biased estimates. They proposed the double bootstrapping truncated regression to replace the Tobit regression. Nowadays decent academic journals do not accept Tobit regressions on efficiency scores. It is strongly suggested that the approach of Simar and Wilson should be followed. Or the authors may use sign-rank tests to avoid the biased estimations by Tobit regressions.

Response 4: Thank you for the reviewer’s detailed comment on this issue. We searched a lot of literatures about Tobit regression model after we got your suggestion. The approach of Tobit may result in biased estimation seriously. So, we adopt your suggestion, upgrade our stata and download the package of simarwilson. Then we re-implement the regression using simarwilson package. Also, we revised the model introduction of regression. Please check it in the line 172 on page 8, in the line 258 on page 13, and table 6. Thank you!

Point 5: The empirical results by Tobit regression are somewhat inconsistent and difficult to explain. For instance, R&D is efficiency-enhancing in one production stage but not in the other. R&D is efficiency-enhancing is some provinces but not in others. It is suggested that direct policy instruments on water consumption efficiency such as fees should be included and empirically tested.

Response 5: Thank you for the reviewer’s detailed comment on this issue. We analysed the explanatory variables and selected 8 variables. And then we re-implemented the regression model using simarwilson package in stata. According to the new outputs from simarwilson package, we found that R&D had consistent influence on industrial water consumption efficiency. Please check it in table 6 on page 14. We also tried to analyse the influence of fees on industrial water consumption. But unfortunately, the data were incomplete. So, we only analysed the indirect policy instruments on water consumption efficiency. Thank you!

Point 6: Many references on pp. 19-21 do not completely show all authors. Some titles of papers are even wrong and do not match the exact full titles of the published articles. They wrong titles may be translated into Chinese and then re-translated into English again. Careful checking and proofreading are necessary before re-submission.

Response 6: Thank you for the reviewer’s detailed comment on this issue. We are aware of this issue, and we checked the references one by one, and added the author’s name that omitted in our manuscript. And we checked the title of references one by one to make sure that the title is correct. Thank you!

Some other revision.

1. We are aware of our limitation in English language, so after we complete the technical revision, we send our manuscript to professional English editing team, and pay for English editing.

2. We reread the manuscript thoroughly, and revised some spelling and language expression, in order to make it more readable.

Thank you for your suggestion for our manuscript.

Round 2

Reviewer 2 Report

1.                  The authors have followed almost all suggestions from this reviewer and made substantial improvements in accordance.

2.                  Since the authors have followed the double-bootstrapping truncated regression of Simar and Wilson (2007) by using the most recent version of Stata, the approach of Simar and Wilson should be further briefly introduced in the text.  In the current version there is no introduction at all for Simar and Wilson’s approach.

3.                  For those Chinese regions with extremely low total-factor water efficiency scores, what kinds of suggestions can be immediately applied to?

Author Response

Response to Reviewer Comments

Point 1: Since the authors have followed the double-bootstrapping truncated regression of Simar and Wilson (2007) by using the most recent version of Stata, the approach of Simar and Wilson should be further briefly introduced in the text. In the current version there is no introduction at all for Simar and Wilson’s approach.

Response 1: Thank you for the reviewer’s detailed comment on this issue. We are aware that there is little introduction about simarwilson approach. So, in this version, we introduced some literatures about simarwilson in introduction sector. Please check it in the line 77-85 on page 2 and page 3. Also, we introduced the principle of simarwilson approach in the methodology sector. In this sector, we also compared the regression methods of Tobit and simarwilson approach, in order to state why we employ simarwilson approach. Please check it in the line 181-197 on page 8. Thank you!

Point 2: For those Chinese regions with extremely low total-factor water efficiency scores, what kinds of suggestions can be immediately applied to?

Response 2: Thank you for the reviewer’s detailed comment on this issue. We are aware that it is important to propose some measures for the provinces with extremely low total-factor water efficiency scores. Meanwhile, we find that the provinces with extremely low total-factor water efficiency scores are IED-dominated regions. In this case, in this version, we proposed some detailed measures for these provinces. Please check it in the line 332-342 on page 16 and page 17. Thank you!

Some other revision.

1. In this revision, we also send our manuscript to professional English editing team, and pay for English editing, after we revised the manuscript.

2. We reread the manuscript thoroughly, and revised some spelling and language expression, in order to make it more readable.

3. We re-typeset the document and add some literatures into the manuscript.

Thank you for your suggestion for our manuscript. Best wishes!
